# Cryostructuring of Polymeric Systems: 62 Preparation and Characterization of Alginate/Chondroitin Sulfate Cryostructurates Loaded with Antimicrobial Substances [note 1]

**DOI:** 10.3390/polym14163271

**Published:** 2022-08-11

**Authors:** Olga I. Vernaya, Andrey N. Ryabev, Tatyana I. Shabatina, Daria L. Karlova, Andrey V. Shabatin, Lyudmila N. Bulatnikova, Alexander M. Semenov, Mikhail Ya. Melnikov, Vladimir I. Lozinsky

**Affiliations:** 1Chemistry Department, M. V. Lomonosov Moscow State University, 119991 Moscow, Russia; 2A. N. Nesmeyanov Institute of Organoelement Compounds, Russian Academy of Sciences, Vavilov Street 28, Bld. 1, 119334 Moscow, Russia; 3N. E. Bauman Moscow State Technical University, 2-nd Baumanskaya 5, 105005 Moscow, Russia; 4A. N. Frumkin Institute of Physical Chemistry and Electrochemistry, Russian Academy of Sciences, Leninsky Ave. 31, Bld. 4, 119071 Moscow, Russia; 5Biology Department, M. V. Lomonosov Moscow State University, 119991 Moscow, Russia

**Keywords:** alginate, chondroitin sulfate, cryostructurates, controlled drug release systems, gentamicin sulfate, dioxidine

## Abstract

Targeted drug release is a significant research focus in the development of drug delivery systems and involves a biocompatible polymeric carrier and certain medicines. Cryostructuring is a suitable approach for the preparation of efficient macroporous carriers for such drug delivery systems. In the current study, the cryogenically structured carriers based on alginate/chondroitin sulfate mixtures were prepared and their physicochemical properties and their ability to absorb/release the bactericides were evaluated. The swelling parameters of the polysaccharide matrix, the amount of the tightly bound water in the polymer and the sulfur content were measured. In addition, FTIR and UV spectroscopy, optical and scanning microscopy, as well as a standard disk diffusion method for determining antibacterial activity were used. It was shown that alginate/chondroitin sulfate concentration and their ratios were significant factors influencing the swelling properties and the porosity of the resultant cryostructurates. It was demonstrated that the presence of chondroitin sulfate in the composition of a polymeric matrix slowed down the release of the aminoglycoside antibiotic gentamicin. In the case of the NH_2_-free bactericide, dioxidine, the release was almost independent of the presence of chondroitin sulfate. This trend was also registered for the antibacterial activity tests against the *Escherichia coli* bacteria, when examining the drug-loaded biopolymeric carriers.

## 1. Introduction

Progress in the development of new drugs is very often associated with the elaboration of fundamentally new dosage forms. In particular, in recent years, numerous pharmaceutical studies have been directed to the formulation of new transport systems of targeted drug delivery and controlled release, which is one of the basic aims in modern pharmacology and nanomedicine. It implies that the drug selectively finds diseased cells in the body and suppresses them without damaging healthy tissues [1]. Controlled drug release medical formulations can solve the problem of maintaining the necessary drug concentration inside an organism and optimize the dosing of used therapeutic medicines thus resulting in the reduction of undesired side effects. The carriers for the controlled drug release systems have to match a large number of requirements. Among them are non-toxicity and biocompatibility, biodegradability, appropriate physicochemical properties, namely rigidity in the dry and wet states, specific porosity, viscoelasticity, swelling dynamics and its extent, etc. Certain biopolymers optimally meet these requirements [2] and are widely used in medicine, the food industry and cosmetology. Some biopolymers find applications as the components of traditional dosage forms, nutritional supplements, implants, coatings for prostheses, topical cosmetics, and so forth. The advantages of these natural macromolecular compounds include high biocompatibility, low toxicity, and environmental friendliness since microorganisms easily decompose such polymeric substrates. Therefore, diverse biopolymers are actively applied in the development of new dosage forms for the drugs’ targeted delivery and controlled-release systems [3,4].

A significant fraction of such biopolymers relates to the non-ionic and ionic polysaccharides [2,5]. Among them are the alginates (**ALG**), well-known and frequently used in pharmacy anionic carbohydrate polymers [6,7,8,9,10]. The natural sources of commercially available alginates are brown algae (*Ascophyllum nodosum, Laminaria Hyperborean, Macrocystis pyrifera*), the cell walls of which contain alginates up to 40% of the dry mass. Alternatively, the biotechnological production of alginates is accomplished with the use of *Azotobacter* and *Pseudomonas* bacteria [8]. The alginates possess such advantages as low toxicity and biocompatibility, easy availability and relatively low cost. All these factors make alginates suitable candidates for precursors to prepare carriers for the drug delivery systems. The alginate-based materials are being extensively explored; they find biomedical applications in tissue engineering [9], drug delivery [3,11], wound dressings [11,12,13]. Alginate-based hydrogels are prospective materials for cell encapsulation and cell delivery in regenerative medicine [14,15]. Alginate films filled with aloe vera extract and silver nanoparticles exhibit antibacterial activity and are promising as antibacterial dressings [16]. Another example of the drug-containing formulation is the alginate-based microgel produced for sustained release of theophylline; the first order kinetic mechanism was shown for drug release from such polysaccharide carriers [17]. Interesting high-porous hydrogels on the basis of chemically cross-linked alginate chains were described by Siboro et al. (2021) [18]. The authors found that the high porosity of the carriers promotes a high loading capacity, and the respective matrices loaded with doxorubicin showed prolonged drug release over ~35 days.

In a view of the approaches to the preparation of macroporous alginate-based matrices, the supercritical drying technique should also be mentioned and needs to be highlighted, which allows the variation in the porosity parameters of such materials over a wide range [19,20]. In turn, one of the efficient approaches to the fabrication of high porous drug carriers is the so-called cryostructuring technique [21,22], which is able to create the macroporous morphology of the resultant polymeric materials via the freezing of the precursor molecular or colloidal solutions, when the polycrystals of frozen solvent perform as porogens. There are two major types of such cryogenically structured matrices [23]: (i) *Cryogels*—the macroporous gels, whose gelation (i.e., the formation of the 3D polymeric network) occurs directly in the frozen system. (ii) *Cryostructurates*—the macroporous materials prepared without the occurrence of the cryotropic gelation process, and the macroporous polymeric matrix is formed by the freezing of the initial liquid system followed by the removal of frozen solvent polycrystals using freeze-drying of cryoextraction techniques. If required, such resultant cryostructurates can be additionally tanned chemically or by irradiation cross-linking.

A similar option has been used for the preparation of the macroporous (sponge-like) biopolymeric drug carriers that were developed in this study.

In addition to imparting specific porosity to the polymer matrix, it is possible to prepare materials of a tailored chemical nature by varying the composition and the concentrations of the precursors, thus influencing the ability of a carrier to absorb and to release the particular drug. In this respect, our attention was focused on the idea of obtaining the polysaccharide carriers that, along with the alginate, would also contain such polysaccharides as chondroitin sulfate (**CNS**). This animal-origin biopolymer, which is a part of cartilage tissue, is a sulfated macromolecular glycosaminoglycan consisting of the alternating monomer units of D-glucuronic acid and sulfated N-acetyl-D-galactosamine [24]. CNS is a biocompatible, low toxic, biodegradable and environmentally friendly biopolymer [25]. Thanks to the presence of sulfate groups, the CNS macromolecules are able to bind the drug molecules, the chemical structure of which includes amino group(s), thus forming ionic pairs with a rather moderate dissociation constant. This feature promotes the slowdown of the subsequent drug release.

In the present study, the cryogenically structured polymeric matrices based on the calcium salts of ALG and CNS polymer mixtures were developed in order to fabricate the carriers for the drug-depot-type formulations that should provide an elongated drug release with respect to the NH_2_-bearing medicinal substance. The antibiotically active drug gentamicin sulfate (**GMS**) (Figure 1a) was employed as a medicine. Gentamicin belongs to the aminoglycoside class of antibiotics that inhibit bacterial protein synthesis and are highly active against aerobic Gram-negative bacteria strains [26]. Therefore, the gentamicin, which is produced by the *Micromonospora purpurea* microbial cells, is widely implemented in medicine and veterinary use. In turn, a broad-spectrum antibacterial NH_2_-free drug dioxidine (**DON**) (Figure 1b) [27] was used in our study for the sake of comparison. It was supposed that GMS could be bound by the CNS through the above-mentioned moderately dissociating salt bridges, while DON could participate only in weak adsorption interactions (mainly H-bonding) with the polysaccharide carrier. Therefore, its release rate would mainly be governed by easier desorption followed by diffusion through the pores of the polymer matrices.

To the best of our knowledge, such cryogenically structured complex Ca(ALG/CNS)-based carriers were unknown earlier; furthermore, their behavior as drug delivery systems has not been reported elsewhere. In this respect, such double-polymer wide-pore polysaccharide materials are novel ones, so their properties and the procedure of their preparation have been, respectively, studied and developed for the first time.

## 2. Materials and Methods

### 2.1. Chemicals

The following substances and reagents were used in the experiments without additional purification: sodium alginate (BDH Chemicals Ltd., Poole, UK) (MM 150 kDa, the viscosity of the 1% aqueous solution 0.63 × Pa·s (23 °C), the content of mannuronic blocks—30%, the content of guluronic blocks—20% and the content of the mixed sequence blocks ~50% [28]); sodium salt of bovine chondroitin sulfate of injectable grade (Bioiberica, Barcelona, Spain); anhydrous calcium chloride (Panreac, Barcelona, Spain); ethanol (96%) (Reakhim, Moscow, Russian Federation); 1% (*w*/*v*) alcohol solution of the Brilliant Green dye (Hippocrates, Moscow, Russian Federation); acetylacetone, phosphoric acid, acetic acid, boric acid, formaldehyde, sodium hydroxide of reagent grade (all from RusChim, Moscow, Russian Federation). Gentamicin sulfate and dioxidine with a purity that meets the requirements of the Russian Pharmacopoeia (Pharmacopoeial requirements 2.1.0043.15, 42-2628-00, 2.2.0014.15, 2.1.0015.15) were purchased from “Mir Farma” (Saratov, Russian Federation). All aqueous solutions were prepared using deionized water.

### 2.2. Preparation of Complex Ca(ALG-CNS) Cryostructurates

Two groups of Na(ALG) + Na(CNS) aqueous solutions of different total concentrations of these biopolymers and their ratio (Table 1) have been prepared:A.The solutions of the same (30 mg/mL) concentration of Na(ALG) and variable (3, 7.5, 15 and 30 mg/mL) concentrations of Na(CNS). In such cases, the mass ratio of these polymers was varied from 1:0.1 to 1:1.B.The solutions of the same (45 mg/mL) total concentration of both polysaccharides and variable Na(ALG)/Na(CNS) mass ratios (from 1:0.1 to 1:1).

The required mixtures of initial dry polymer powders were dissolved in water with stirring (700 rpm) at room temperature for 3.5 h until the homogeneous solutions were obtained. These solutions were dosed in the 2.5 g portions into plastic Petri dishes with an inner diameter of 38 mm (OAO Medpolimer, Moscow, Russian Federation). Then the samples were placed in the chamber of a Proline RP 1840 ultracryostat (Lauda, Königshofen, Germany) for freezing, stored here at –25 °C for 18 h and then lyophilized for 18 h in a vacuum (0.04 mBar) using an Alpha 1-2 LD plus freeze-drier (Martin Christ, Osterode am Harz, Germany). The resulting dry disks were immersed in a saturated ethanolic solution of CaCl_2_, kept for 16 h at room temperature with periodic stirring. Thereafter, the disks were rinsed three times with ethanol; the absence of calcium salt in the final portion of ethanol was checked by the evaporation test. These rinsed disks were then dried in a vacuum desiccator over the calcined CaCl_2_ granules. The Ca(ALG) cryostructurates without CNS additives (9 in Table 1) were fabricated similarly using 30 mg/mL Na(ALG) solution; the swelling parameters and the spongy morphology of the latter cryostructurates have been reported previously [29].

### 2.3. Loading of Ca(ALG/CNS) Cryostructurates with GMS or DON

Loading of cryogenically structured polymeric carriers with the model drugs (gentamicin sulfate, dioxidine) was performed according to the procedure described elsewhere [30,31]. In brief, the Ca(ALG-CNS)-based cryostructurates were loaded by their soaking in the solutions of GMS or DON, then frozen at −20 °C, freeze-dried using a VirTis Wizard 2.0 instrument (SP Industries, Stone Ridge, USA) for 24 h (temperature range: on a condenser from −42 to −55 °C; on the product from −30 to +30 °C, residual pressure in the chamber (6–12) × 10^−2^ Torr). The concentrations of GMS in the solutions used for loading in the spongy polysaccharide carriers were as follows: 3.74% (for the cryostructurates 1 and 5, Table 1), 2.96% (for the samples 2 and 6, Table 1), 2.26% (for the carriers 3 and 7, Table 1), 1.87% (for the sponges 4 and 8) and 4.0% in the case of CNS-free matrix (9, Table 1). These values of GMS concentration were calculated in such a way as to maintain an equal molar ratio of the drug to the sulfo-groups of CNS. Upon loading the respective spongy carriers with DON, its concentrations were 4 times lower than those of GMS since the water solubility of dioxidine is quite low; however, high molar extinction coefficient of this substance allowed its subsequent photometric quantification even in dilute solutions.

### 2.4. Characterization

#### 2.4.1. Characterization of Dry Ca(ALG/CNS) Cryostructurates

The amount of bound water (K. Fischer coulometric method) [32] and the sulfur content were analyzed using standard procedures in the Microanalysis Laboratory of the Institute of the corresponding author.

#### 2.4.2. Optical Microscopy

The macroporous morphology of the dry and water-swollen Ca(ALG/CNS) cryostructurates contrasted with Brilliant Green was studied according to the previously described technique [29] using an SMZ1000 optical stereomicroscope (Nikon, Tokyo, Japan) equipped with an MMC-50C-M digital imaging system (MMCSoft, Saint Petersburg, Russian Federation).

#### 2.4.3. Scanning Electron Microscopy

SEM images of the dried polymeric samples were recorded using a scanning electron microscope FEI QUANTA 650 FEG (Thermo Fisher Scientific, Hillsboro, OR, USA) installed in the Collective Facilities Center of A.N. Frumkin Institute of Physical Chemistry and Electrochemistry RAS. The images were recorded at 5 kV accelerating voltage using secondary electrons detector; the examined polymeric material was not additionally coated.

#### 2.4.4. FTIR-Spectroscopy

The IR spectra of the samples were registered on a Tensor II (Bruker GmbH, Mannheim, Germany) FTIR spectrometer equipped with an attenuated total reflection (ATR) module A225/Q platinum.

#### 2.4.5. Swelling Degree of Ca(ALG/CNS) Cryostructurates in Water

Swelling characteristics of the polymeric samples were studied using the gravimetric procedure. The freeze-dried Ca(ALG/CNS) disk of known weight (*m*_dry_) was immersed in 50 mL of water and kept there for 3 h with occasional stirring. Then the swollen sample was transferred to a glass Petri dish, free water was carefully removed from the surfaces of the disk by blotting with filter paper, and the sample was weighed, thus determining the total mass (*m*_swn_) of the swollen spongy material. After that, this sample was placed on a glass Schott filter, and a plastic Petri dish containing a 100 g kettlebell was installed on the top of swollen sponge. Under this load the capillary liquid was removed from the sponge in a vacuum (~15 mm Hg) for 1 min as described previously [29]. Further, this pressed-off wet disk was weighed in order to determine the mass of the swollen polymeric phase (*m*_wet_).

Using the measured values, the following swelling characteristics of the Ca(ALG/CNS) cryostructurates were calculated:

Total swelling capacity (*S*_tot_) of the biopolymer material in water:*S*_tot_ = *m*_swn_:*m*_dry_ (g of bound + free water/g of dry polymer)(1)

Specific amount of free water (*M*_afw_) in the capillaries of the sponge sample:*M*_afw_ = (*m*_swn_ − *m*_wet_):*m*_dry_ (g of free water/g of dry polymer)(2)

Swelling degree (*S*_pol_) of the polymer framework:*S*_pol_ = (*m*_wet_ − *m*_dry_):*m*_dry_ (g of bound water/g of dry polymer)(3)

#### 2.4.6. Kinetic Curves for the Drug Release

The kinetic curves for the drug release were recorded using a UV–Visible-NIR double beam Jasco V-770 spectrophotometer (JASCO Corporation, Tokyo, Japan). The concentration of GMS released from the carriers was measured in accordance with the Russian Technical Standard #5951-91. A buffer solution (pH 2.56) was prepared by adding 150 mL of the 1.0 M sodium hydroxide solution to 1000 mL of a mixture of 0.2 M orthophosphoric, 0.2 M acetic, and 0.2 M boric acids. Then, 4 mL of acetylacetone and 10 mL of the formaldehyde aqueous solution (37% *w*/*w*) were added to 136 mL of the buffer thus resulting to the “analytical solution”. A weighed piece (~2 mg) of the dry GMS-bearing carrier was placed in a beaker with 25 mL of physiological saline and incubated with periodical shaking. The 100 µL portions of liquid phase were taken at certain time intervals and added to test tubes each containing a mixture of 2.5 mL of distilled water and 2.5 mL of the “analytical solution”. The tube was immersed in a water bath (100 °C) for 15 min, cooled to room temperature, and optical absorption of the solution at 345 nm was measured.

In the case of DON release, the pieces of the drug-loaded cryostructurates of equal weight were placed singly in each beaker with 50 mL of saline and incubated with periodical shaking. The 1 mL portions of liquid phase were taken at certain time intervals, the optical absorption at 375 nm of these solutions was measured, after which each portion of the solution was returned to the respective “mother” sample.

All data were presented as mean with standard deviation (±SD). For statistical analysis, the Excel T-TEST function was used, and statistically significant differences were described as the *p*-values less than 0.05 (*p* < 0.05).

#### 2.4.7. Characterization of Antibacterial Activity

The evaluation of the antibacterial activity of the drug-loaded carriers was performed by the disk-diffusion procedure similar to that earlier described used for other drug-loaded cryogenically structured biopolymeric carriers [30,31,33]. The experiments were carried out using the *E. coli* bacterial strain from the collection of the Microbiology Department, Biology Faculty of M.V. Lomonosov Moscow State University. Standard Petri dishes that contained 20 mL of agar nutrient medium dried for 24 h (thickness of the medium layer was 4 mm) were used. The bacterial cells of the test culture in an amount of 10^8^ were seeded onto each 90 mm agar dish, after which the drug-loaded carrier as the disk of 4 mm in diameter and 2 mm in height was placed onto the centre of the agar layer. The measurements of the growth inhibition zones (**GIZ**) for this bacterial strain were carried out after 24 h of incubation, and then every 24 h, the GIZ size was checked for changes. Statistically reliable results were obtained by a nine-fold repetition of the measurements for each series of samples.

## 3. Results and Discussion

### 3.1. Ca (ALG/CNS) Cryostructurates

Complex double-polymer Ca(ALG/CNS) spongy cryostructurates of different total concentrations and mass ratios of two anionic polysaccharides were fabricated according to the scheme shown in Figure 2 and described in Section 2.2. The process included the preparation of a mixed aqueous solution of Na-salts of these macromolecular precursors, i.e., Na(ALG) and Na(CNS), its freezing, the removal of crystallized ice by sublimation, followed by the treatment of the dry wide-pore cryostructurates with an alcoholic solution of calcium chloride to convert the carboxylate groups of alginate and the sulfate groups of chondroitin sulfate into the form of weakly dissociated calcium salts, that are poorly water soluble.

The freezing of the initial polymer solutions and their subsequent freeze-drying resulted in the formation of the sponge-like cryostructurates. A characteristic feature of such polymer matrices is the presence of a system of interconnected macropores. This texture is formed in the resultant material after the sublimation of the frozen solvent [21,22,23,34].

The geometrical parameters, weight, and density of the thus-prepared dry Ca(ALG/CNS)-cryostructurates formed as disks, as well as their sulfur content, are summarized in Table 2. Since the sublimation of ice from frozen aqueous solutions of hydrophilic polymers does not lead to an exhaustive removal of water, some water remains firmly bound to the polymer matrix. Therefore, the amount of such bound water in our lyophilized samples was determined by the K. Fischer method (Section 2.2) and the values so-determined were taken into account during subsequent calculations of the data given in Table 2.

As it was assumed, with an increase in the total concentration of Na(ALG) and Na(CNS) in the initial solution of the polymeric precursors (group A, Table 1 and Table 2), the mass of the corresponding Ca(ALG/CNS) disks and their specific density also increased. In the case of samples of group B (constant concentration of both polysaccharides in the initial solutions), the density of the resultant cryostructurates slightly increased with the increasing content of chondroitin sulfate. At the same time, the geometric dimensions of the lyophilized samples varied insignificantly. Thus, it is clear that an increase in the proportion of Na(CNS) in the initial solution of these polymers should lead to the growth of the sulfur content in the final cryostructurates (groups A and B, Table 2).

Figure 3 presents the examples of optical microscope images of such polysaccharide sponges in the dry state. Their macroporous morphology is typical for similar freeze-dried materials, quite heterogeneous in terms of the texture and size of the structural elements. The extended white elements in these microphotographs are the thin polymeric walls of macropores with an overall width of 10–20 µm. Dark areas belong to macropores, their sizes lie over the range from 100 to 500 and more µm. An increase in the total concentration of polymer components in their initial solution from 33 to 60 mg/mL (samples 1–4, 1) resulted in some decrease in the average cross-section of macropores.

The interconnected character of wide pores in the dry cryostructurates (with the pore cross-section from tens to hundreds µm) is also well visualized in the SEM micrographs (Figure 4). The pore size of these samples varied in a wide range. The SEM results additionally confirmed the possibility of controlling the average pore size of such cryostructurates by varying the concentration of polysaccharides in the initial solution. For instance, the average pore cross-section in sample (1) (total concentration of polymers in the feed solution—33 mg/mL; Table 1) was 62 µm, that in sample (4) (total concentration of polymers in the feed solution—60 mg/mL; Table 1) was 157 µm.

The swelling parameters of hydrophilic polymeric materials in aqueous media are important characteristics that are especially necessary for the correct application of such systems in practice, in particular as materials of biomedical interest [2,3,35,36,37]. Therefore, in the present study, the values of the total swelling capacity (*S*_tot_), the specific amount of free water (*M*_afw_) inside the volume of capillary size macropores, and the swelling degree (*S*_pol_) of the polymer framework (the walls of the macropores of these spongy materials) for the Ca(ALG/CNS) cryostructurates 1–8 (Table 2) were found (for the procedure, see Section 2.4.5). The results of such measurements are summarized in Table 3.

These experimental results show that with an increase in the total concentration of both polysaccharides, as well as in the content of chondroitin sulfate in the initial solution (group A, Table 1 and Table 3) the values *S*_tot_ and *M*_afw_ of the respective Ca(ALG/CNS) cryostructurates decreased. At the same time, the swelling degree of the polymeric phase (*S*_pol_) inside such spongy materials increased markedly. The latter fact clearly indicates an increase in the hydrophilicity of the walls of the macropores with an increase in the share of sulfate-containing polysaccharide. This trend was also observed for group B (Table 1 and Table 3), but in the latter case, it was less noticeable. Hence, the higher the ability of the polymeric walls of macropores in such cryostructurates to swell, the smaller the internal space available for filling with a free liquid in the capillaries (at least the narrowest of them). Therefore, the values of the *S*_tot_ and *M*_afw_ should decrease; this trend is indeed observed for the cryogenically structured polysaccharide sponges under discussion (Table 3).

Some features of the macroporous morphology of water-swollen Ca(ALG/CNS) cryostructurates (group A, Table 1, Table 2 and Table 3) are illustrated by the microphotographs in Figure 5. Since the walls of the macropores of such spongy materials after their swelling in aqueous media are virtually transparent and poorly distinguishable under the microscope, the corresponding samples were contrasted by a short-term treatment with the solution of the Brilliant Green. The dye was well adsorbed by the polymer phase and intensively stained the pore walls. Thus, in the micrographs in Figure 5, the walls of the macropores were dark, and the macropores themselves were lighter. It turned out that the macropores of the swollen samples were somewhat larger in comparison with the corresponding dry samples (Figure 3). This is a consequence of the fact that upon swelling, the volume of the material increases. The relationship between the size of the macropores and the total concentration of alginate and chondroitin sulfate in the initial solution of these polysaccharides is the same as that for the dry samples, namely, when the concentration of polymeric precursors increases, the average pore diameter decreases. Therefore, one can draw the conclusion that the variation of the Na(ALG) and Na(CNS) total concentration and their ratios in the precursor solution allow more or less adjustment to the osmotic properties (Table 3) and the microstructure (Figure 2, Figure 3 and Figure 4) of the resultant Ca(ALG/CNS) cryostructurates.

### 3.2. FTIR-Spectra of the Ca(ALG/CNS) Cryostructurates

Certain information on the complex Ca(ALG/CNS) cryostructurates was obtained from their FTIR spectra (Figure 6). Thus, in the spectrum of the CNS-free cryogenically structured Ca(ALG) sponge (Figure 6a) there were the bands in the region of 3300 cm^−1^ inherent in the stretching vibrations of OH-groups, and at 2929 cm^−1^ belonging to CH-groups. There were also the bands of asymmetric and symmetric stretching vibrations of COO^−^ (1591 and 1404 cm^−1^), COH (1295 cm^−1^), COC, CC, CO (1083 cm^−1^) and CC, COC (1024 cm^−1^) structural elements. In the case of cryostructurates that contained not only Ca(ALG), but also Ca(CNS), the bands characteristic of the chondroitin-sulfate were additionally observed in the spectrum (Figure 6b), namely, Amide I (1634 cm^−1^), Amide II (1560 cm^−1^), O-SO_3_H vibrations [38] (1226 cm^−1^) and planar vibrations of CO-NH (996 cm^−1^). Thus, these data confirm the presence of CNS macromolecules in the composition of the complex polysaccharide cryostructurates prepared according to the processing scheme shown in Figure 1.

### 3.3. Ca(ALG/CNS Cryostructurates Loaded with GMS or DON

As was pointed out in the “Introduction”, we consider the above-described Ca(ALG/CNS)-cryostructurates as potential carriers for the depot-forms of the NH_2_-bearing drug molecules. Therefore, such polysaccharide sponges were further loaded with the aminoglycoside antibiotics gentamicin sulfate (GMS) and, for the sake of comparison, with the NH_2_-free bactericide substance dioxidine (DON).

The incorporation of the drug components into the polymer matrices was performed by the method of the cryochemical drug modification [39]. To this end, the sample of the Ca(ALG/CNS) sponge was immersed in the drug solution, then frozen and subjected to freeze-drying (for details see Section 2.3). As is known [39,40,41], similar cryogenic treatment of the water-dissolved drugs allows the formation of nanoparticulate matter and, in some cases, is able to change the crystal structure of the drugs to form their new polymorphs [42]. As a result, such micronization of the drug particles and changing of their crystal structure give rise to the improvement of drug solubility, bioavailability and therapeutic efficiency [39,42].

Loading the Ca(ALG/CNS) cryostructurates with DON or GMS caused the expected changes in the FTIR spectra of the respective drug-containing sponges. In the case of the carrier loaded with dioxidine (Figure 7a), along with the bands belonging to the alginate and chondroitin sulfate, the bands characteristic of DON were present in the spectrum. The latter bands were as follows: 1510 cm^−1^ (bending vibrations of the quinoxaline ring), 1376 cm^−1^ (stretching vibrations of NO), 1322 cm^−1^ (COH), 160 and 1119 cm^−1^ (CH), 1054 cm^−1^ (COH), and 962 cm^−1^ (bending vibrations of NO) [36]. Since the NH_2_-free DON molecules are simply absorbed via non-ionic interactions (mainly, H-bonding) by the polysaccharide framework of the carrier without the formation of salt bridges, the band at 1230 cm^−1^ of the O-SO_3_H vibration inherent in the CNS (Figure 6b), was also observed in the respective spectrum (Figure 7a). In turn, in the spectrum of the GMS-bearing carrier (Figure 7b), the band of stretching O-SO_3_H vibrations was virtually absent, thus indicating the ionic interaction of NH_2_-containing GMS molecules with sulfo-groups of chondroitin sulfate. With that, the band at 1525 cm^−1^ characteristic of the NH deformation vibrations belonged to this drug. In turn, the bands around 1024 cm^−1^ (CC, COC and COH vibrations) became wider in comparison to the spectrum of the drug-free polysaccharide carriers (Figure 6a).

In addition, some visual information on the structural features of our drug-loaded carriers was obtained from the SEM data. First, loading of the sponges with DON or GMS did not cause noticeable changes in the pore size (Figure 8a). Second, the size of the small drug particles located on the inner surfaces of the pore walls of the GMS-loaded Ca(ALG/CNS)-cryostructurate turned out to be over the range of 50–200 nm (Figure 8b). Therefore, the subsequent release of the respective drug upon placing the DON- or GMS-containing sponges in aqueous medium would include the dissolution of such small drug particles.

### 3.4. Features of GMS and DON Release from the Drug-Loaded Ca(ALG/CNS) Cryostructurates

The kinetic curves for the release of GMS from the drug-loaded carriers are shown in Figure 9, where m_t_ is the amount of the released substance for the time t, and m is the amount of the initially loaded drug. It was found that the sensible changes of the released drug amount from the CNS-free Ca(ALG)-carrier were virtually completed in 50–60 min (Figure 9a). At the same time, in the case of GMS release from the CNS-containing drug-loaded carriers, a significant elongation of the release time took place (Figure 9b–e). Therefore, the higher the fraction of CNS in the composition of a particular Ca(ALG/CNS) cryostructurate, the slower the releasing process for this NH_2_-containing drug, i.e., gentamicin. For instance, in the case of GMS-loaded sponge with the highest content of the CNS within the polymeric phase (4; Table 1 and Table 3), the release time was about 7 h. For all samples, no increase in the GMS concentration in the outer liquid occurred after the 7 h incubation period under the experimental conditions used. When these samples were further stored in the capped tubes for 10 days, no increase in the concentration of released GMS was detected either. In other words, the pseudo-equilibrium state between the absorbed and desorbed drug in these systems had been reached.

The FTIR spectrum of the dried spongy carrier after the above-discussed GMS release experiment is depicted in Figure 10. The comparison of this spectrum with that for the GMS-loaded cryostructurate before drug release (Figure 7b) shows that in the former case it was the decrease in the intensity of GMS-related bands at 1525 cm^−1^ and 607 cm^−1^ (bending vibrations of NH and SO_2_, respectively). This fact obviously testifies that a small fraction of GMS still remains bound under experimental conditions, most probably in the ionic manner, through the salt bridges with the sulfate group of CNS.

As for the DON release from the respective drug-loaded polysaccharide carriers, the following pattern was observed (Figure 11). In the case of the DON-loaded Ca(ALG) carrier, the concentration of dioxidine in the outer liquid medium stopped changing at 50–60 min (Figure 11a), i.e., absolutely analogously to the gentamicin release in the case of the similar CNS-free cryostructurate. These results obviously point to the absence of some specific interactions of both GMS and DON with the Ca-alginate matter. In turn, if for the NH_2_-containing GMS molecules, a certain deceleration of their release from the Ca(ALG/CNS)-carriers was observed (Figure 9b–e), a similar effect of the slowdown of the release of the NH_2_-free DON molecules from the CNS-containing sponges was considerably less pronounced (Figure 11b–e). Namely, the time of the pseudo-equilibrium state reached in the latter cases was no more than 1.5–2 h and did not markedly depend on the chondroitin sulfate content in the carriers consisting of these two polysaccharides. From our viewpoint, a rather obvious reason for such differences in the release behavior of GMS and DON is the formation of poorly dissociating salt bonds between the sulfate groups of CNS and amino-groups of GMS, whereas in the DON molecules such groups were absent, and the drug desorption was not additionally hindered.

### 3.5. Antibacterial Properties of the GMS- or DON-Loaded Ca(ALG/CNS) Cryostructurates

The so-called disk diffusion procedure has been applied for the evaluation of antimicrobial properties of the drug-loaded polysaccharide cryostructurates (see Section 2.4.7). The *E coli* bacteria were implemented as the test microorganisms, and the GMS- or DON-loaded sponges of the compositions 1–4 and 9 (Table 1) were used as the drug carriers in these experiments. The same sponges, but without the antibacterial substances, as well as the disks of filter paper impregnated with 0.3 mg/mL solutions of GMS or DON were used as the reference samples. The diameter of the “growth inhibition zone” (**GIZ**) was measured and considered as the indicator of the antibacterial activity. Such zones are formed onto the surface of solid nutritional agar due to the bactericide release from the respective carrier toward the microbial mat [30,31,33]. The quantitative results of the respective tests are summarized in Table 4, and the appearance of the typical Petri dish with the tested samples is demonstrated by the photograph in Figure 12.

The data of Table 4 testify that the drug-free Ca(ALG) and Ca(ALG/CNS) spongy carriers themselves, as well as the pure filter paper, did not, as expected, exhibit any antibacterial activity. In turn, around the drug-loaded discs of filter paper, there were the widest growth inhibition zones, since the used antibacterial substances were almost not retained by the paper material. In the case of the GMS- or DON-loaded cryostructurates, the size of GIZs was decreased by ~3%, i.e., was a little lower, most probably because such cryogenically structured sponges consisting of the ionic polysaccharides were able to retain somewhat both drugs via the additional, not very strong, absorption interactions. When testing the antibacterial activity of the GMS-loaded carriers, the samples with an equal content of the bactericide were used. Therefore, an increase in the content of CNS in the matrix resulted in some decrease in the observed antibacterial activity because of a decrease in the rate of drug release. Upon the comparison of the GMS-loaded CNS-containing and CNS-free cryostructurates (respectively, samples 2–4 and 9 in Table 4) it was found that statistically significant differences (*p* < 0.05) in the values of GIZ were in the case of samples 4 and 9. The differences at the level of statistical trend (0 < *p* < 0.1) against the CNS-free sample 9 were observed for the samples 2 and 3.

In turn, upon testing the DON-containing sponges, the GIZ values were very similar for all samples, regardless of the CNS content. In this case, the level of statistical significance in all cases exceeded 0.1, which confirms the absence of statistically significant differences.

Such results correlated well with the data of release kinetics (Figure 9 and Figure 11) indicating a certain retaining ability by the CNS-containing carriers with respect to the NH_2_-bearing drug, gentamicin, and the virtual absence of that ability with respect to the NH_2_-free molecules of dioxidine.

## 4. Conclusions

Currently, various ionic polysaccharides are widely employed in various fields including their use as materials of biomedical interest, in particular, as the polymeric matrices for drug delivery. In this study, one novel example of similar materials, namely, the spongy cryostructurates based on the calcium salt from the mixture of two ionic polysaccharides, alginate (ALG) and chondroitin sulfate (CNS), were prepared, characterized and their potential to act as drug carriers was evaluated. Medical substances, gentamicin sulfate (GMS) and dioxidine (DON), were used as bactericides, in which the molecules of the former have the amino groups, and the molecules of the latter do not. The fabrication of the target polymeric drug carriers included the preparation of the aqueous solutions of the (Na)ALG and (Na)CNS mixtures, their freezing, freeze-drying, ionotropic cross-linking by the treatment with a saturated solution of CaCl_2_ in ethanol, rinsing and vacuum drying. Subsequent immersion of the thus-prepared spongy materials in the GMS- or DON-containing aqueous solutions allowed the loading of the biopolymeric carriers with the bactericide. The examination of the carrier properties showed that an increase in the total concentration of both polysaccharides and an increase in the CNS content in the initial solutions resulted in a decrease in the water absorbing capacity and the specific content of free moisture in the volume of the cryostructurate macropores. At the same time, the swelling extent of the polymer matter of the pore walls increased markedly. Optical microscopy and SEM data testified to the interconnected wide-pore morphology of such cryogenically structured polysaccharide drug carriers. Subsequent kinetic studies of the GMS and DON released from the drug-loaded sponges revealed a longer release period for the former substance in comparison to that of the latter one, thus pointing to a certain retention of the NH_2_-containing GMS by the CNS in the composition of the carrier. Most probably, it occurred owing to the formation of the relatively stable salt bridges between amino- and sulfate groups of the drug and the carrier, respectively. Therefore, in view of possible biomedical applications as the vehicles of amino-containing drugs, the more preferable types of such double-polymer matrices are the Ca(ALG/CNS)-cryostructurates with an increased content of the chondroitin component. The disk-diffusion procedure with the *E coli* bacteria strain was applied to determine the antibacterial effects of the drug-loaded Ca(ALG/CNS) carriers. These tests demonstrated that such materials potentially could serve as efficient drug-delivery systems.

## Figures and Tables

**Figure 1 polymers-14-03271-f001:**
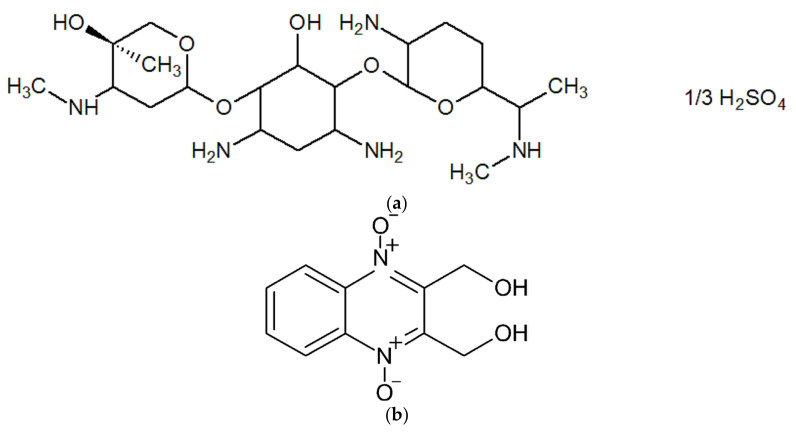
Chemical structures of gentamicin sulfate (**a**) and dioxidine (**b**).

**Figure 2 polymers-14-03271-f002:**
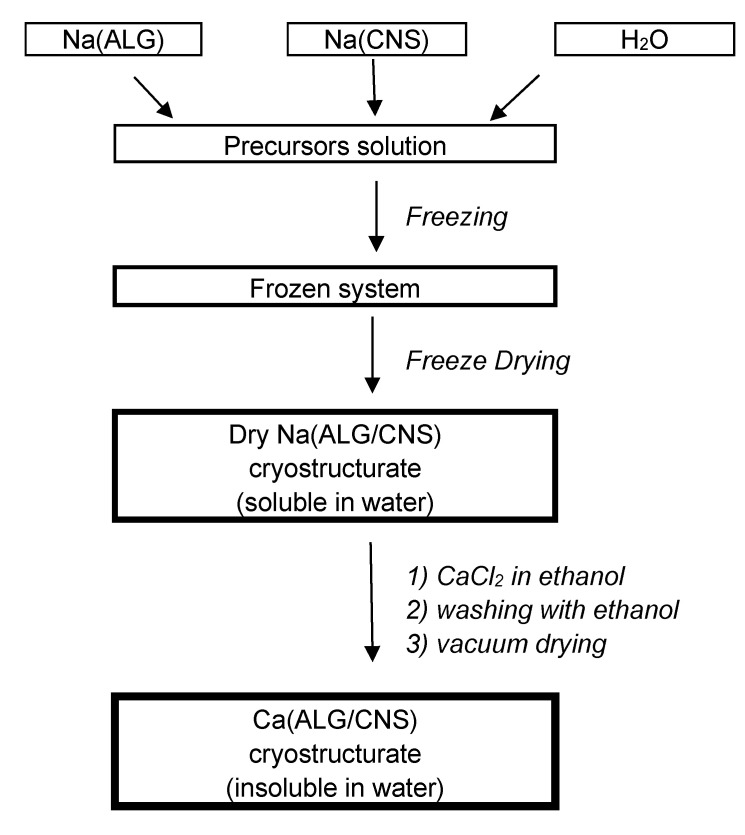
General scheme for the preparation of Ca(ALG/CNS) cryostructurates.

**Figure 3 polymers-14-03271-f003:**
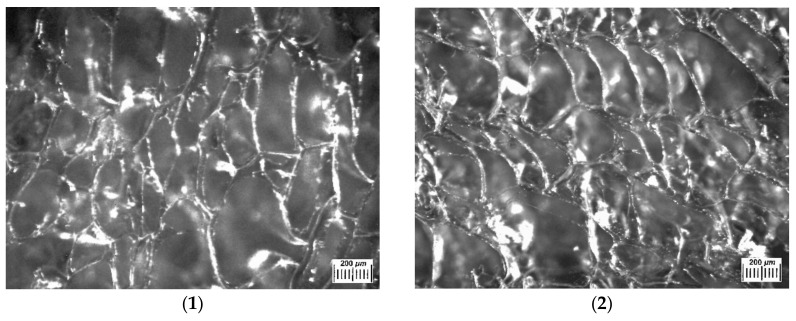
Microphotographs obtained as images in reflected light by means of an optical stereomicroscope for the dry Ca(ALG/CNS) cryostructurates (numbering of the images corresponds to the samples **1**–**4** in Table 1 and Table 2).

**Figure 4 polymers-14-03271-f004:**
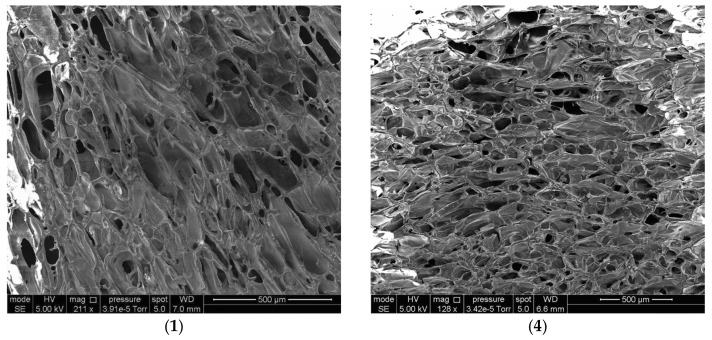
SEM micrographs of dry Ca(ALG/CNS) cryostructurates (numbering of the images corresponds to samples **1** and **4** in Table 1 and Table 2).

**Figure 5 polymers-14-03271-f005:**
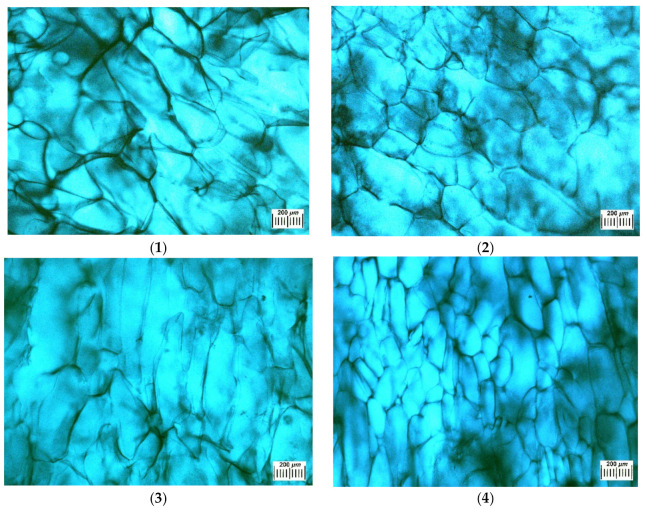
Microstructure of Ca(ALG/CNS) cryostructurates swollen in water and contrasted with a Brilliant Green solution (micrographs obtained using an optical stereomicroscope; numbering of the images corresponds to the samples **1**–**4** in Table 2).

**Figure 6 polymers-14-03271-f006:**
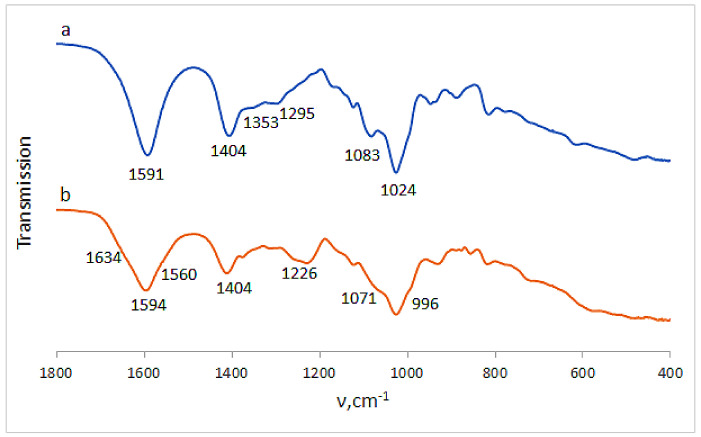
FTIR spectra of the Ca(ALG) (**a**) (9 in Table 1) and Ca(ALG/CNS) (**b**) (4 in Table 1 and Table 3) cryostructurates.

**Figure 7 polymers-14-03271-f007:**
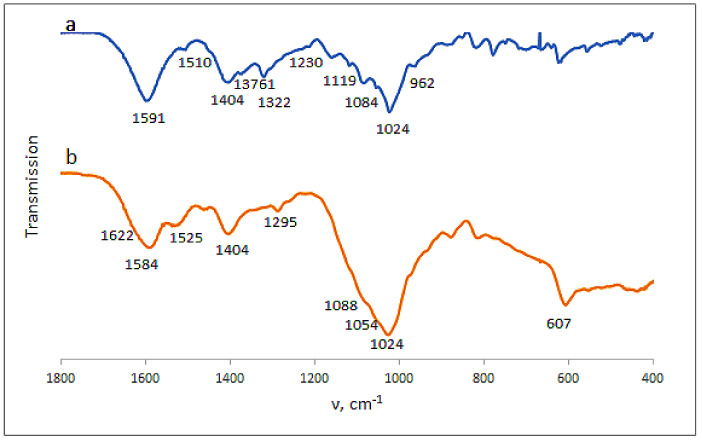
FTIR spectra of the Ca(ALG/CNS) cryostructurates (4 in Table 1 and Table 3) loaded with GMS (**a**) and DON (**b**).

**Figure 8 polymers-14-03271-f008:**
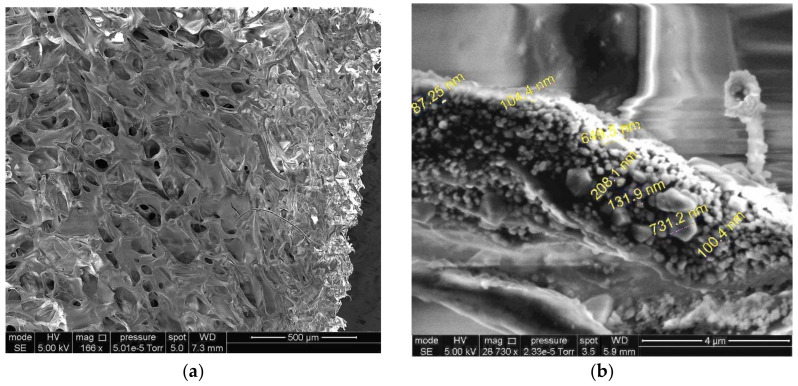
SEM images at small (**a**) and large (**b**) magnification of the dry Ca(ALG/CNS) cryostructurate (sample 4, Table 1) loaded with GMS.

**Figure 9 polymers-14-03271-f009:**
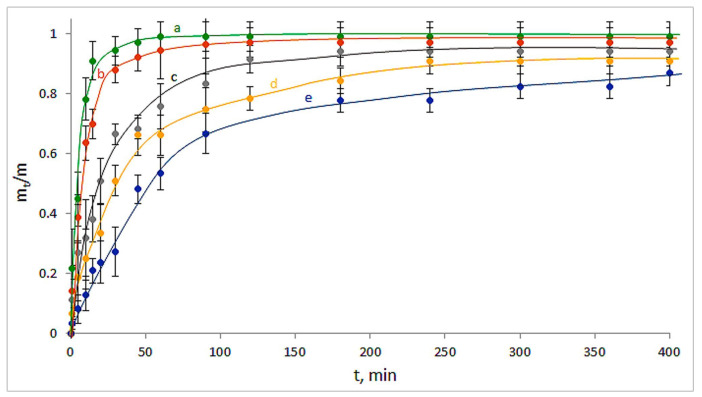
Kinetic curves for the GMS release from the drug-loaded carriers: Ca(ALG) (**a**—9 in Table 1), Ca(ALG/CNS) (**b**–**e**—1–4 in Table 1 and Table 3).

**Figure 10 polymers-14-03271-f010:**
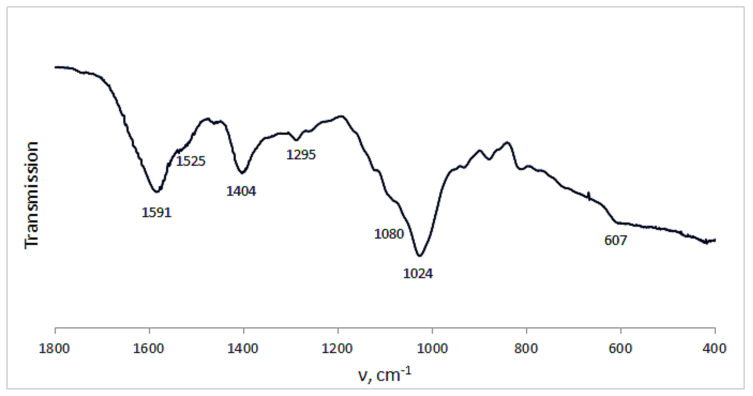
FTIR spectrum of the Ca(ALG/CNS) cryostructurate (4 in Table 1 and Table 3) after the release of preliminary loaded GMS.

**Figure 11 polymers-14-03271-f011:**
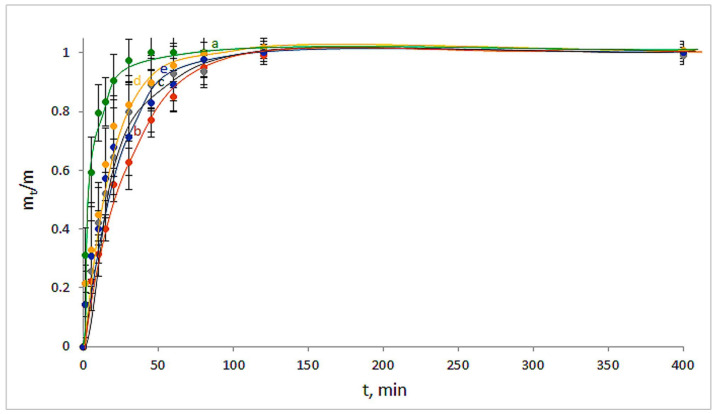
Kinetic curves for the DON release from the drug-loaded carriers: Ca(ALG) (**a**—9 in Table 1), Ca(ALG/CNS) (**b**–**e**—1–4 in Table 1 and Table 3).

**Figure 12 polymers-14-03271-f012:**
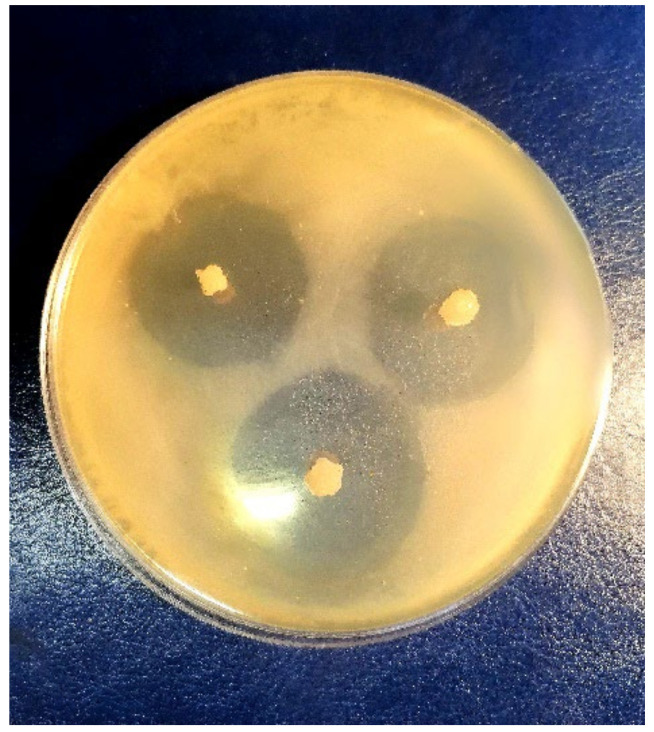
Growth-inhibition zones for the *E. coli* cells around the DON loaded Ca(ALG/CNS) spongy carrier (2 in Table 1) after 24 h of incubation.

**Table 1 polymers-14-03271-t001:** Composition of the initial solutions used for the preparation of complex Ca(ALG/CNS) and Ca(ALG) cryostructurates.

Sample	Group	Composition of the Initial Solutions			
		Na(ALG) Concentration (mg/mL)	Na(CNS) Concentration (mg/mL)	Total Concentration Na(ALG) and Na(CNS) (mg/mL)	Mass Ratio Na(ALG)/Na(CNS)
1	A	30	3	33	1:0.1
2		30	7.5	37.5	1:0.25
3		30	15	45	1:0.5
4		30	30	60	1:1
5	B	40.9	4.1	45	1:0.1
6		36	9	45	1:0.25
7 (3)		30	15	45	1:0.5
8		22.5	22.5	45	1:1
9	-	30	-	30	1:0

**Table 2 polymers-14-03271-t002:** Parameters of dry Ca(ALG/CNS) cryostructurates.

Sample	Group	Disk Diameter (mm)	Disk Thickness (mm)	Disk Weight * (g)	Disk Density * (g/cm^3^)	Sulfur Content * (wt%)
1	A	35.2 ± 0.1	2.40 ± 0.10	0.0765 ± 0.0011	0.0328 ± 0.0020	0.57 ± 0.10
2		35.2 ± 0.1	2.20 ± 0.10	0.0856 ± 0.0014	0.0400 ± 0.0022	1.00 ± 0.10
3		35.4 ± 0.1	2.35 ± 0.15	0.1023 ± 0.0013	0.0452 ± 0.0024	1.52 ± 0.14
4		35.3 ± 0.2	2.30 ± 0.10	0.1174 ± 0.0016	0.0553 ± 0.0015	2.91 ± 0.25
5	B	35.2 ± 0.1	2.30 ± 0.10	0.1014 ± 0.0030	0.0448 ± 0.0034	0.50 ± 0.10
6		34.8 ± 0.1	2.40 ± 0.10	0.1024 ± 0.0032	0.0451 ± 0.0031	0.91 ± 0.15
7 (3)		35.4 ± 0.1	2.35 ± 0.15	0.1023 ± 0.0013	0.0456 ± 0.0024	1.52 ± 0.14
8		35.2 ± 0.2	2.30 ± 0.10	0.1041 ± 0.0015	0.0465 ± 0.0027	2.87 ± 0.25

* Without the bound water content (determined by the K. Fischer method).

**Table 3 polymers-14-03271-t003:** Swelling parameters of Ca(ALG/CNS) cryostructurates.

Sample	Group	Swelling Parameters of Ca(ALG/CNS) Cryostructurates		
		*S*_tot_ (g of Bound + Free Water/g of Dry Polymer)	*M*_afw_ (g of Free Water/g of Dry Polymer)	*S*_pol_ * (g of Bound Water/g of Dry Polymer)
1	A	69.3 ± 4.9	62.5 ± 3.9	5.8 ± 0.3
2		59.2 ± 3.6	50.6 ± 2.8	7.0 ± 0.4
3		39.7 ± 1.9	30.9 ± 1.7	7.9 ± 0.5
4		26.2 ± 1.0	16.4 ± 0.8	8.9 ± 0.7
5	B	89.1 ± 6.9	81.1 ± 6.3	7.0 ± 0.2
6		61.2 ± 5.0	52.9 ± 4.7	7.3 ± 0.3
7 (3)		39.7 ± 1.9	30.7 ± 1.7	7.9 ± 0.5
8		26.5 ± 1.3	18.3 ± 0.7	8.2 ± 0.4

* Polymer weight—without the content of bound water (determined by the K. Fischer method).

**Table 4 polymers-14-03271-t004:** Growth-inhibition zones for the *E. coli* bacteria around the filter paper disks and the polysaccharide cryostructurates loaded with GMS and DON.

Composition of the Carrier ^a^	Loaded Antibacterial Substance	GIZ (mm)
1	GMS	34.1 ± 1.2
2		33.8 ± 1.2
3		33.6 ± 1.3
4		33.2 ± 1.2
9		35.1 ± 1.5
1	DON	33.9 ± 1.1
2		33.3 ± 1.6
3		33.5 ± 1.2
4		34.0 ± 1.8
9		33.7 ± 1.7
1	none	0
2		0
3		0
4		0
9		0
Filter paper	none	0
	GMS	35.2 ± 1.6
	DON	35.4 ± 0.8

^a^ The numbers correspond to the samples in Table 1.

## Data Availability

Not applicable.

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
