# Peer review of "Cryostructuring of Polymeric Systems: 62 Preparation and Characterization of Alginate/Chondroitin Sulfate Cryostructurates Loaded with Antimicrobial Substances†"

_polymers, 2022, doi:10.3390/polym14163271_

Round 1

Reviewer 1 Report

The subject of finding novel drug delivery systems is of high interest nowadays. In this context, the authors of this manuscript have prepared macroporous matrices for antimicrobial drugs by cryostructuration of a mixture of two anionic polysaccharides,  sodium alginate (NaALG) and sodium chondroitin sulfate (NaCNS). Just, some weak points need to be remediated to increase the clarity of the paper and the access of the readers to its content. 

1.    The authors announced in lines 145-150 that two groups of cryostructurates based on NaALG and NaCNS have been prepared: A. The solutions of the same (30 mg/mL) concentration of Na(ALG) and variable 147 (3, 7.5, 15 and 30 mg/mL) concentration of Na(CNS); B. The solutions of the same (45 mg/mL) total concentration of both polysaccha-149 rides and variable Na(ALG)/Na(CNS) mass ratio (from 1:0.1 to 1:1), but in Table 1 the reader can see that the Na(ALG)/Na(CNS) mass ratio has been variable also in the series A. Please, adjust the presentation!

2.    After preparation, the cryostructurates have been rinsed only with ethanol (lines 159-161). Why the authors did not use water for the removal of soluble species as much as the cryostructurates are stabilized by ionic crosslinks with Ca2+?

3.    Lines 172-176, the authors should explain why they have used different concentrations of drugs to load the cryostructurates.

4.    The resolution of microphotographs in Figures 3, 4 and 8 must be improved.

 Editorial mistakes:

Line 91, “It is the similar option has been used…” must be “A similar option has been used…”.

Line 96, “In this respect…” instead of “It this respect…”.

Line 108, “… provide an elongated …”

Author Response

Reviewer-1

Comments and Suggestions for Authors

The subject of finding novel drug delivery systems is of high interest nowadays. In this context, the authors of this manuscript have prepared macroporous matrices for antimicrobial drugs by cryostructuration of a mixture of two anionic polysaccharides,  sodium alginate (NaALG) and sodium chondroitin sulfate (NaCNS). Just, some weak points need to be remediated to increase the clarity of the paper and the access of the readers to its content. 

  1. The authors announced in lines 145-150 that two groups of cryostructurates based on NaALG and NaCNS have been prepared: A. The solutions of the same (30 mg/mL) concentration of Na(ALG) and variable 147 (3, 7.5, 15 and 30 mg/mL) concentration of Na(CNS); B. The solutions of the same (45 mg/mL) total concentration of both polysaccha-149 rides and variable Na(ALG)/Na(CNS) mass ratio (from 1:0.1 to 1:1), but in Table 1 the reader can see that the Na(ALG)/Na(CNS) mass ratio has been variable also in the series A. Please, adjust the presentation!

Answer – The following additional sentence (in blue) has been inserted in the revised manuscript:

  1. The solutions of the same (30 mg/mL) concentration of Na(ALG) and variable (3, 7.5, 15 and 30 mg/mL) concentration of Na(CNS). In such case the mass ratio of these polymers was varied from 1:0.1 to 1:1.

  1. After preparation, the cryostructurates have been rinsed only with ethanol (lines 159-161). Why the authors did not use water for the removal of soluble species as much as the cryostructurates are stabilized by ionic crosslinks with Ca2+?

Answer – The following additional text (in blue) has been inserted in the revised manuscript:

Thereafter, the disks were rinsed three times with ethanol; the absence of calcium salt in the final portion of ethanol was checked by the evaporation test. Thus rinsed disks were then dried in a vacuum desiccator over the calcined CaCl2 granules.

  1. Lines 172-176, the authors should explain why they have used different concentrations of drugs to load the cryostructurates.

Answer – The following additional text (in blue) has been inserted in the revised manuscript:

The concentrations of GMS in the solutions used for loading in the spongy polysaccharide carriers were as follows: 3.74 % (for the cryostructurates 1 and 5, Table 1), 2.96% (for the samples 2 and 6, Table 1), 2.26% (for the carriers 3 and 7, Table 1), 1.87% (for the sponges 4 and 8) and 4.0% in the case of CNS-free matrix (9, Table 1). These values of GMS concentration were calculated in such a way as to maintain an equal molar ratio of the drug to the sulfo-groups of CNS.

  1. The resolution of microphotographs in Figures 3, 4 and 8 must be improved.

Answer – In accordance to the reviewer’s recommendation the resolution of these images have been increased.

Reviewer 2 Report

This manuscript describes the manufacturing of alginate/chondroitin cryostructurates loaded with two types of antibacterial drugs, characterizations, and activity testing. The manuscript is generally well organized and clearly presented. You could include more references of existing studies (excluding further self-citations). You should focus more on what is the novelty of this experiment,  especially in the Results and Discussion part. 

Specific comments:

line 73: “...were described in the paper [15].” –add first author and year

line 74: “in connection with which... “–rephrase

line 110: "Gentamicins" in Gentamicin (singular)

line 171-172: “ -42...-55°C and - 30...+30°C “– without dots

table 4, chapter 3.5: include the statistic with significant differences between the results, and then comment on the results in the further discussion, since it seems like the results are not statistically different

4. conclusions: you should drive some conclusions about the different compositions you used for cryostructurates

Author Response

Reviewer-2

Comments and Suggestions for Authors

This manuscript describes the manufacturing of alginate/chondroitin cryostructurates loaded with two types of antibacterial drugs, characterizations, and activity testing. The manuscript is generally well organized and clearly presented. You could include more references of existing studies (excluding further self-citations). You should focus more on what is the novelty of this experiment,  especially in the Results and Discussion part. 

Answer – The following additional text (in blue) with respect of the novelty has been inserted in the revised manuscript:

To the best of our knowledge, such cryogenically-structured complex Ca(ALG/CNS)-based carriers were unknown earlier, as well as their behavior as the drug delivery systems was not reported elsewhere. In this respect, such double-polymer wide-pore polysaccharide materials are the novel ones, so as their properties and the procedure of their preparation have been, respectively, studied and developed for the first time.

Specific comments:

line 73: “...were described in the paper [15].” –add first author and year

done

line 74: “in connection with which... “– rephrase

The phrase has been changed as follows:

The authors found that high porosity of the carriers promotes high loading capacity, and the respective matrices loaded with doxorubicin showed the prolonged drug release during ~35 days.

line 110: "Gentamicins" in Gentamicin (singular)

done

line 171-172: “ -42...-55°C and - 30...+30°C “– without dots

done

table 4, chapter 3.5: include the statistic with significant differences between the results, and then comment on the results in the further discussion, since it seems like the results are not statistically different

Answer-1 – The following paragraph (in blue) has been added to the section 2.4.6:

All data were presented as mean with standard deviation (±SD). For statistical analysis the Excel T-TEST function was used, and statistically significant differences were described as the p-values less than 0.05 (p<0.05).

Answer-2 – The changes (in blue) in the section 3.5 of the revised manuscript are as follows:

The data of Table 4 testify that the drug-free Ca(ALG) and Ca(ALG/CNS) spongy carriers themselves, as well as the pure filter paper, did not, as expected, exhibit any antibacterial activity. In turn, around the drug-loaded discs of filter paper there were the widest growth inhibition zones, since the used antibacterial substances were not almost retained by the paper material. In the case of the GMS- or DON-loaded cryostructurates, the size of GIZs was decreased by ~3%, i.e. was a little lower, most probably because such cryogenically-structured sponges consisting of the ionic polysaccharides were able to retain somewhat both drugs via the additional, not very strong, absorption interactions. When testing the antibacterial activity of the GMS-loaded carriers, the samples with an equal content of the bactericide were used. Therefore, an increase in the content of CNS in the matrix resulted in some decrease in the observed antibacterial activity because of a decrease in the rate of drug release. Upon the comparison of the GMS-loaded CNS-containing and CNS-free cryostructurates (respectively, samples 2-4 and 9 in Table 4) it was found that statistically significant differences (p<0.05) in the values of GIZ were in the case of samples 4 and 9. The differences at the level of statistical trend (0<p<0.1) against the CNS-free sample 9 were observed for the samples 2 and 3.

In turn, upon testing the DON-containing sponges, the GIZ values were very similar for all samples, regardless of the CNS content. In this case, the level of statistical significance in all cases exceeded 0.1, which confirms the absence of statistically significant differences.

  1. conclusions: you should drive some conclusions about the different compositions you used for cryostructurates

The following phrase (in blue) has been added to the ‘Conclusions’ section:

Therefore, in a view of possible biomedical application as the vehicles of amino-containing drugs, the more preferable types of such double-polymer matrices are the Ca(ALG/CNS)-cryostructurates with increased content of chondroitin component.
